# Homer1 Protects against Retinal Ganglion Cell Pyroptosis by Inhibiting Endoplasmic Reticulum Stress-Associated TXNIP/NLRP3 Inflammasome Activation after Middle Cerebral Artery Occlusion-Induced Retinal Ischemia

**DOI:** 10.3390/ijms242316811

**Published:** 2023-11-27

**Authors:** Weihao Lv, Xiuquan Wu, Yanan Dou, Yiwen Yan, Leiying Chen, Zhou Fei, Fei Fei

**Affiliations:** 1Department of Neurosurgery, Xijing Hospital, Air Force Medical University, No. 127, Changle West Road, Xincheng District, Xi’an 710032, China; weihaoneuro@163.com (W.L.); wuxiuquan2023@163.com (X.W.); douyanan@163.com (Y.D.); yanyiw2023@163.com (Y.Y.); 2Department of Ophthalmology, Xijing Hospital, Air Force Medical University, No. 127, Changle West Road, Xincheng District, Xi’an 710032, China

**Keywords:** MCAO-induced retinal ischemia, RGCs, pyroptosis, Homer1, ER stress, AMPK

## Abstract

Retinal ischemia, after cerebral ischemia, is an easily overlooked pathophysiological problem in which inflammation is considered to play an important role. Pyroptosis is a kind of cell death pattern accompanied by inflammation. Homer scaffold protein 1 (Homer1) has anti-inflammation properties and protects against ischemic injury. However, little is known about pyroptosis following middle cerebral artery occlusion (MCAO)-induced retinal ischemia and the regulatory mechanisms involved by Homer1 for the development of pyroptosis. In the present study, retinal ischemic injury was induced in mice by permanent MCAO in vivo, and retinal ganglion cells (RGCs) were subjected to Oxygen and Glucose Deprivation (OGD) to establish an in vitro model. It was shown that TXNIP/NLRP3-mediated pyroptosis was located predominantly in RGCs, which gradually increased after retinal ischemia and peaked at 24 h after retinal ischemia. Interestingly, the RGCs pyroptosis occurred not only in the cell body but also in the axon. Notably, the occurrence of pyroptosis coincided with the change of Homer1 expression in the retina after retinal ischemia and Homer1 also co-localized with RGCs. It was demonstrated that overexpression of Homer1 not only alleviated RGCs pyroptosis and inhibited the release of pro-inflammatory factors but also led to the increase in phosphorylation of AMPK, inhibition of ER stress, and preservation of visual function after retinal ischemia. In conclusion, it was suggested that Homer1 may protect against MCAO-induced retinal ischemia and RGCs pyroptosis by inhibiting endoplasmic reticulum stress-associated TXNIP/NLRP3 inflammasome activation after MCAO-induced retinal ischemia.

## 1. Introduction

Stroke is currently the main factor affecting human health and quality of life, with ischemic stroke accounting for approximately 87% of all stroke-related events [1,2,3]. In clinical diagnosis and treatment, it has been found that some patients with acute ischemic stroke often experience ipsilateral retinal ischemia, which often leads to acute painless loss of monocular vision [4,5]. The main anatomical reason is that the ophthalmic artery and internal carotid artery are adjacent to each other. Moreover, in animal experiments, it was found that the successful establishment of a model of middle cerebral artery occlusion can also be accompanied by ischemia of the ipsilateral retina [6,7], leading to the phenomenon of “whitening” of the ipsilateral eyeball [8]. Insufficient blood supply to the ophthalmic artery can cause an infarction of the retinal ganglion cells (RGCs) and optic nerve [9,10], causing visual disturbances and blindness [11,12,13].

It is well-known that after ischemia, activation of the Nod-like receptor family pyrin domain-containing 3 (NLRP3) inflammasome converts pro-caspase-1 into mature caspase-1, which then activates gasderminD (GSDMD), resulting in pyroptosis of the cells, accompanied by the release of interleukin-18 (IL-18) and interleukin-1β (IL-1β). On the other hand, ischemia can also cause an imbalance in cellular redox status. The thioredoxin-interacting protein (TXNIP) not only maintains redox balance but also interacts with NLRP3 to trigger inflammatory processes [14,15,16]. In addition, the interaction between TXNIP and NLRP3 can also cause GSDMD-mediated cell death, ultimately forming the TXNIP/NLRP3-mediated cell pyroptosis signal axis [14]. And TXNIP may also connect organelle stress, such as endoplasmic reticulum (ER) stress, to NLRP3 inflammasome activation [17]. Furthermore, ER stress can cause the interaction between downstream TXNIP and NLRP3, leading to the activation of NLRP3 inflammasomes, thereby increasing the expression of various pro-inflammatory factors and inducing pyroptosis [18,19,20].

Homer scaffold protein 1 (Homer1), as a postsynaptic scaffold protein, is an important component of the nervous system [21]. It is reported that Homer1 is involved in the pathogenesis of cerebral hemorrhage [22], cerebral ischemia [23], and traumatic brain injury [24] through various pathways, such as inhibiting neuroinflammation. However, as an extension of the central nervous system-retina, the impact of Homer1 on the endoplasmic reticulum stress-associated TXNIP/NLRP3-mediated cell pyroptosis after the ischemic injury is not yet known.

## 2. Results

### 2.1. Generation of MCAO-Induced Retinal Ischemia Model

In the experiments, none of the sham-operated mice died. The mortality did not significantly differ among the experimental MCAO groups. No significant adverse effects were observed in adeno-associated virus (AAV)-infected experimental MCAO groups. Figure 1A shows our experimental schedule. 2, 3, 5-triphenyl tetrazolium chloride stain (TTC) staining (Figure 1B) confirmed that the MCAO model was successfully generated, accompanied by the whitening of the right retinal (Figure 1C). The results of retinal oscillatory potentials (OPs) showed that the amplitude of OPs in the retinal ischemia group was significantly reduced or even disappeared compared with that of the sham group (Figure 1D), demonstrating that the MCAO-induced retinal ischemia model was successfully generated.

### 2.2. TXNIP/NLRP3-Mediated Pyroptosis Was Activated after Retinal Ischemia

Firstly, a Western blot was performed to assess the expression of TXNIP/NLRP3-mediated pyroptosis-related protein (TXNIP/NLRP3/Cl-capase1/GSDMD/IL-1β/IL-18) of the ischemic retina at different time points. The results showed that the level of pyroptosis-related protein initially increased and then decreased, with a peak at 24 h (Figure 2A–G). Then, immunofluorescence-staining found that in the whole layer of the retina, the pyroptosis cells were mainly RGCs (Figure 2H), and it was noteworthy that pyroptosis occurred not just in the RGC bodies but also in axons (marked by the white arrow). In addition, compared with that of the Con, the expression level of TXNIP/NLRP3-mediated pyroptosis-related protein was at a peak at 6 h after Oxygen and Glucose Deprivation (OGD) in vitro (Figure 2I–O).

Through in vivo and in vitro experiments, it was shown that the expression of pyroptosis-related proteins was up-regulated after retinal ischemia, and pyroptosis occurred mainly in RGCs.

### 2.3. Occurrence of Pyroptosis Coincided with the Change of Homer1 Expression in the Retina after Retinal Ischemia

To verify the Homer1 temporal induction expression trend in the retina, Western blot (WB) was used to detect the expression trend of Homer1 after retinal ischemia (Figure 3A,B). The expression of Homer1 protein also increased and peaked at 24 h, which was surprisingly consistent with the time change trend of pyroptosis. Additionally, the change in the trend of the Homer1 transcription level was consistent with that of the protein level (Figure 3C). Furthermore, the immunofluorescence result showed that Homer1 was localized in RGCs (Figure 3D).

OGD experiments were conducted on primary RGCs in vitro to simulate the microenvironment of retinal ischemia in vivo. After OGD treatment, the protein (Figure 3E,F) and mRNA level (Figure 3G) of Homer1 continued to increase, and the expression level reached its peak at 6 h after OGD.

In short, it was found that the time change trend of Homer1 was consistent with the pathological process of RGCs pyroptosis.

### 2.4. Overexpression of Homer1 Alleviated RGC Pyroptosis after Retinal Ischemia

To investigate the role of Homer1 in the process of retinal ischemia, the MCAO-induced retinal ischemia was induced 4 weeks after AAV injection. WB showed that Homer1-OE significantly improved TXNIP/NLRP3-mediated pyroptosis (Figure 4A–F). Immunofluorescence showed that the proportion of pyroptosis cells in Homer1-OE was lower than that of the retinal ischemia (I) group (Figure 4G,H). ELISA results showed that Homer1-OE effectively reduced the expression of IL-18 (Figure 4I) and IL-1β (Figure 4J), whereas Homer1-KD exacerbated the inflammatory reaction.

In addition, RGCs were transduced through lentivirus, resulting in Homer1-OE and Homer1-KD. WB results showed that Homer1-OE significantly decreased the expression of the protein of the TXNIP/NLRP3 signal axis (Figure 5A–F). ELISA displayed that Homer1-OE inhibited the release of IL-18 and IL-1β (Figure 5G,H). In addition, flow cytometry results indicated that Homer1-OE reduced the proportion of GSDMD-positive RGCs (Figure 5I,J). At the same time, Homer1-KD aggravated the expression of characteristic proteins of the pyroptosis signaling pathway and the release of inflammatory factors. These data illustrated that Homer1 may play a protective role by inhibiting the pathological process of TXNIP/NLRP3-mediated pyroptosis.

### 2.5. Overexpression of Homer1 Inhibited ER Stress by Regulating AMPK Activity after Retinal Ischemia

With the gradual deepening of the cognition of endoplasmic reticulum stress, it is gradually believed that CCAAT-enhancer-binding protein homologous protein (CHOP) and binding immunoglobulin (Bip) are markers of the endoplasmic reticulum stress response [25]. The results of animal experiments showed that ER stress-related protein expression was significantly lower in mice within the Homer1-OE group than that of the I group. In contrast, related protein expression was significantly higher in mice with the Homer1-KD than that of the I group (Figure 6A–F). These data illustrated that Homer1 may play a protective role after MCAO-induced retinal ischemia by inhibiting ER stress.

AMPK has been reported to be the key factor in regulating the inflammatory response and ER stress [26]. And Liver kinase B1 (LKB1) is the main upstream kinase of AMPK, promoting its phosphorylation in response to energy stress [27]. Therefore, we further detected p-AMPK and p-LKB1 expression levels in vitro and in vivo, and WB results suggested that Homer1-OE promoted the AMPK phosphorylation by regulating the phosphorylation of LKB1, and Homer1-KD inhibited this alteration (Figure 7A–D). In addition, there is no direct interaction between Homer1 and AMPK, as demonstrated by immunoprecipitation in both retinas of the ischemic and normal groups (Appendix A). These results indicated that Homer1 regulated the phosphorylation of AMPK through indirect regulation.

### 2.6. Overexpression of Homer1 Preserved Visual Function after Retinal Ischemia

The hematoxylin and eosin (HE) staining of retinal cross-sections showed the decreased thickness of retinal tissue 24 h after retinal ischemia, and this effect was markedly inhibited with Homer1-OE treatment, while Homer1-KD further exacerbated tissue damage. The thickness of the retina (marked by the yellow arrow) in the sham, I, Homer1-KD, and Homer1-OE groups were 206.9 ± 7.458, 175.6 ± 3.502, 128.5 ± 1.951, and 190.4 ± 5.529 μm, respectively (Figure 8A,B). Flash visually evoked potentials (FVEPs) were measured 24 h after retinal ischemia. The P2 amplitudes in the sham, I, Homer1-KD, and Homer1-OE groups were 33.6 ± 1.735, 17.57 ± 1.266, 13.30 ± 1.323, and 25.87 ± 1.804 μV, respectively. Homer1-OE increased P2 amplitudes, whereas these alternations were prevented by Homer1-KD, indicating that Homer1-OE suppressed deterioration of damage (Figure 8C,D).

## 3. Discussion

In the present study, it was shown that TXNIP/NLRP3-mediated pyroptosis increased gradually and peaked 24 h after MCAO-induced retinal ischemia. The pyroptosis occurred predominantly in RGCs and was located not only in the cell body but also in the axon. With the development of the pyroptosis, Homer1 co-localized with RGCs and peaked 24 h after retinal ischemia. Overexpression of Homer1 could inhibit ER stress-related TXNIP/NLRP3-mediated pyroptosis by regulating the AMPK signaling pathway, which in turn improved the visual function of the retina after ischemia.

Retinal ischemia and cerebral ischemia concurrence is a small-probability event in which inflammation is considered to play an important role, and it can be easily overlooked [28]. Pyroptosis is a kind of cell death pattern accompanied by inflammation, characterized by the formation of non-selective pores in the cell membrane [29], and is increasingly investigated in ischemic stroke [30]. It has been reported that inflammasome complexes are activated to subsequently induce pyroptosis and cause the release of inflammatory factors into the extracellular fluid, which in turn enhances tissue inflammation [31]. However, inflammation-induced pyroptosis following MCAO-induced retinal ischemia and the possible regulatory mechanisms are poorly understood. Interestingly, it was found for the first time in our study that after retinal ischemia, the pyroptosis of RGCs occurred not only in the cell body but also in axons. It is speculated that the activated GSDMD protein may move to nerve axons via potential trafficking in the cytoplasm, triggering axonal damage.

The mechanism of cell pyroptosis caused by the TXNIP/NLRP3 signal axis has been reported in other diseases, such as intestinal ischemia-reperfusion injury [14], cadmium-induced liver injury [32], and LPS-induced acute lung injury [33]. In this study, it was demonstrated that TXNIP/NLRP3-mediated pyroptosis also played an important role after retinal ischemia. Based on the temporal pattern of pyroptosis, we used immunofluorescence staining to detect cells that experienced pyroptosis at 24 h of retinal ischemia. For the first time, it was found that the characteristic molecule GSDMD of pyroptosis was mainly expressed in the innermost layer of the retina (i.e., the optic ganglion cell layer). When Neuronal β-III tubulin antibody was used to label RGCs, it was found that in the entire layer of the retina, the pyroptosis cells were mainly RGCs.

An increasing number of studies have shown that ER stress plays an important role in ischemic injury [34,35]. Hypoxia has been shown to cause abnormal protein folding in neurons, leading to the accumulation of misfolded proteins in the ER, which aggravates ER stress and results in ultimately irreversible neurological deficits [36]. It is reported that ER stress causes Bip chaperones to dissociate from ER stress receptors, activating the receptors [37], and ER stress induces the activation of the CHOP [38,39]. In our study, Bip and CHOP proteins as markers of ER stress changed accordingly with the regulation of Homer1 in this experiment, verifying the role of ER stress in the TXNIP/NLRP3-mediated pyroptosis-signaling pathway after MCAO-induced retinal ischemia. However, ER stress has not been explored in depth, which is one of the shortcomings of this study.

It is reported that the activation of AMPK leads to the phosphorylation and degradation of TXNIP, which can further inhibit ER stress and lower the level of NLRP3 [40,41,42]. Most studies suggest that the phosphorylation of AMPK provides neuroprotection after an acute stroke. For example, increased levels of AMPK following a stroke can inhibit microglial activation and reduce neutrophil infiltration into brain tissue [43], and AMPK attenuates hippocampal glutamate neurotoxicity by inhibiting ER stress-mediated TXNIP/NLRP3 inflammasomes [40]. In this study, the protective effect of the AMPK signaling pathway in pyroptosis signaling after retinal ischemia was confirmed. In addition, as it has also been shown that Homer1 plays a protective role through the AMPK signaling pathway in the HT22 neuronal oxidative stress model [44], our study found that the overexpression of Homer1 significantly increased AMPK phosphorylation levels and inhibited TXNIP/NLRP3-mediated pyroptosis.

Ischemic stroke remains a severely disabling and fatal disease worldwide, and some patients suffer from complications such as visual impairment. In population-based studies, asymptomatic retinal emboli developed in 0.32% to 2.9% of the population. The incidence of retinal emboli in patients with acute stroke is more than 10 times higher than that in large studies based on the general population [6]. In our study, great attention was paid to the condition in which retinal ischemia occurred at the same time as cerebral ischemia, which was truly simulated by our MCAO–retinal ischemia. Clinically, a large area of cerebral infarction resulting from occlusion of the middle cerebral artery can cause patients’ disturbance of consciousness. At this time, the clinician’s focus is on how to restore the perfusion of the ischemic hemisphere in the time window, and without doubt, this is not a good thing for the ipsilateral ischemic retina. It is proposed from our study that for patients with disturbed consciousness and massive cerebral infarction, retinal function tests on the ischemic side should be supplemented simultaneously to avoid the loss of monocular vision and alleviate neurological deficits in patients.

Although it was found in this study that Homer1 can regulate ER stress-related TXNIP/NLRP3-mediated pyroptosis through the AMPK signaling pathway in the MCAO–retinal ischemia model, some limitations cannot be ignored. First, Homer1 could exert protective effects against retinal ischemia in a variety of ways. However, only AMPK-dependent pathways were validated in this study. Second, only the anti-inflammatory and antioxidant effects of Homer1 were evaluated in this study, and the role of Homer1 in apoptosis or autophagy was not investigated further. Third, retinal ganglion cells, glial cells, cones, rods, and other types of cells in the retina are inevitably damaged by ischemia and hypoxia after retinal ischemia. However, in this study, we found that pyroptosis was scattered throughout the retinal layers and found that the cells that experienced pyroptosis were mainly retinal ganglion cells. Therefore, we speculate that there may be multiple pathways of regulation, such as apoptosis/pyroptosis in the retina after ischemia. Therefore, further studies are needed to be carried out to investigate other mechanisms of action of Homer1 in retinal ischemia.

## 4. Materials and Methods

### 4.1. Animals

All animal experiments were performed in accordance with protocols approved by the Institutional Ethics Committee of Xijing Hospital. All experimental procedures were approved by the Institutional Animal Care and Use Committee of Air Force Medical University (approval No. IACUC-20220630). C57BL/6 male mice (6–8 weeks) weighing 25–30 g were purchased from the Shanghai Model Organisms Center, Inc. (Shanghai, China). All animals were maintained in a temperature-controlled facility with 12 h of light/dark cycle at 23 ± 3 °C and 30–70% humidity.

### 4.2. Lentivirus and Adeno-Associated Virus

Lentiviruses and AAVs were designed by Hanbio Biotechnology Co., Ltd. Primary RGCs were infected with lentivirus to stably overexpress Homer1 (Homer1-OE) or knock down Homer1 (Homer1-KD). The lentiviral information is as follows: Homer1-OE: GV358/Ubi-MCS-3FLAG-SV40-EGFP-IRES-puromycin; Homer1-KD: GV248/hU6-MCS-Ubiquitin-EGFP-IRES-puromycin.

The intravitreal injection was performed with a Hamilton Micro Needle injector (Hamilton^®^ Microlite™ Syringe, volume 5 μL. needle L 43 mm; Sigma Aldrich, St Louis, MO, USA). Briefly, ref. [45] exposed the superior nasal region of anesthetized mice eyes; after disinfecting the periorbital area and ocular surface, the injection is performed between the corneal limbus and the equatorial region of the eye and towards the posterior pole of the eye. Waiting for a complete 1 min after injection, the removal of the injector and the slow application of levofloxacin hydrochloride eye gel was performed on mice eyes for the prevention of infection. The intravitreal injection was performed only in the right eye of the mouse. The AAV information is as follows: Homer1-OE: AAV2/DJ-pHBAAV-CMV-Homer1-3flag-T2A-ZsGreen; Homer 1-KD: AAV2/DJ-pHBAAV-U6-Homer1shRNA-CMV-EGFP. The targeting sequence of the siRNA was 5′-GCATTGCCATTTCCACATA-3, and the transcript sequence for overexpressing Homer1 was NM_011982.

### 4.3. Experimental Design

A total of 72 C57/BL mice were used, and all animal experiments were designed based on the 3R principle. In the first step, the expression of TXNIP/NLRP3-mediated pyroptosis-related protein was assessed in the sham group and at different time points. Mice were randomly divided into six groups: sham, 2 h, 4 h, 8 h, 24 h, 48 h. The retinal tissues from each group were collected for Western blot analysis (n = 6/each group). The cellular localization of GSDMD was detected using double immunofluorescence staining (sham and 24 h groups, n = 6/each group).

In the second step, to study the effects of Homer1, mice were randomly divided into sham, retinal ischemia (I), Homer1-knock down (Homer1-KD), and Homer1-overexpression (Homer1-OE). Each group (n = 6) of animals was euthanized 24 h after retinal ischemia. The right ischemic retina of six mice in each group was used for WB, ELISA assays, and Flash visually evoked potentials (FVEPs) (n = 6/each group). The right ischemic retina of six mice in each group was used to obtain frozen sections and for immunohistochemistry (n = 6/each group).

In vitro, RGCs were divided into control group (Con, RGCs cultured in normal Neurobasal) and OGD groups (RGCs cultured in glucose-free Neurobasal, and then cultured at 37 °C in an OGD chamber containing 5% CO_2_ and 95% N_2_). In the OGD group, RGCs were randomly divided into six groups at OGD 30 min, 1 h, 2 h, 4 h, and 6 h. In addition, RGCs infected with Homer1-OE or KD lentiviruses (Hanbio Biotechnology Co., Ltd., Shanghai, China) were divided into the Homer1-OE group and the Homer1-KD group. All in vitro experiments were independently repeated three times, and in each of these three biological replicates, three technical replicates were made.

### 4.4. Induction of Permanent MCAO-Induced Retinal Ischemia

MCAO surgery was performed with minor modifications to the previous description [46]. Briefly, animals were initially anesthetized by inhalation of 5% isoflurane (in N_2_/O_2_ 70%/30% mixture), and sedation was maintained by inhalation of 2% isoflurane. Oxygen saturation (SpO_2_) was analyzed and maintained at 90% using a pulse oximeter (SurgiVet, Model V3304; Waukesha, WI, USA). Body temperature was monitored by the rectal probe and maintained between 36.5 °C and 37.5 °C with a heating lamp. Next, the incision site of the neck skin was disinfected with iodine complex, and a midline neck incision was made to separate the right common carotid artery, right external carotid artery, and right internal carotid artery. Silicone-coated nylon microwires (L2000, Guangzhou Jialing Biotechnology Co., Ltd., Guangzhou, China) were inserted into the right internal carotid artery through a right external carotid artery incision to occlude the right middle cerebral artery. Sham mice were operated identically, except that the middle segment of the right carotid artery was not occluded.

### 4.5. Isolation of Primary Cultured Retinal Ganglion Cells (RGCs)

Primary RGC cells were extracted and identified as previously described [47]. Briefly, Primary RGCs were isolated from postnatal three-day (P3) C57BL/6 mice pups. The eyes were enucleated, and the retina was mechanically dissected. The whole retinas were incubated in papain solution (16.5 U/mL) for 30 min. Afterward, macrophages and endothelial cells in the suspension were washed and cleared by antimacrophage antiserum. RGCs were specifically bound to the panning plates containing anti-Thy1.2 antibody, and unbound retinal cells were removed by washing with DPBS. Purified RGCs were released by trypsin incubation and grown in Neurobasal/B27 media (Invitrogen, Carlsbad, CA, USA).

### 4.6. Oxygen and Glucose Deprivation (OGD)

To mimic the ischemic injury, the culture medium was renewed with glucose-free Neurobasal after being washed three times with phosphate-buffered saline (PBS), and the RGCs were then cultured in a specific chamber containing 5% CO_2_ and 95% N_2_ at 37 °C to simulate ischemic injury.

### 4.7. 2, 3, 5-Triphenyl Tetrazolium Chloride Stain (TTC)

Mice were sacrificed, and the brains were cut into four 2 mm thick slices. Sections were stained with 1% 2, 3, 5-triphenyl tetrazolium chloride (TTC) (Sigma, St. Louis, MO, USA) for 10 min at 37 °C and fixed in 10% PFA overnight. Metabolically active tissue reduces TTC to form a red product, while stroke tissue remains white because its metabolic enzymes are compromised. The images were taken by a digital camera.

### 4.8. Immunofluorescence Staining of Retinal Sections

Immunohistochemical analysis was performed as previously described [48]. Mice were anesthetized at specific time points after permanent MCAO-induced retinal ischemia. Then, the eyes were enucleated and fixed in 4% PFA for 1 day, dehydrated, and frozen into sections. After that, frozen sections were blocked with 5% bovine serum albumin (BSA) and 0.5% Triton X-100 in PBS. The following antibodies were used: mouse anti-Neuronal βIII-Tublin (1:100; Abcam, Cambridge, UK), rabbit anti-Homer1 (1:100; Abcam, Cambridge, UK), rabbit anti-GSDMD-N (1:100; Affinity, MI, USA), donkey anti-rabbit IgG (H + L) highly cross-adsorbed secondary antibody, Alexa Fluor Plus 488 (1:1000; Invitrogen, Carlsbad, CA, USA), and donkey anti-mouse IgG (H + L) highly cross-adsorbed secondary antibody, Alexa Fluor Plus 555 (1:1000; Invitrogen, Carlsbad, CA, USA). The GSDMD-N positive RGCs were measured in four adjacent areas within 1 mm of the optic disc. In the in vivo experiment, five visual fields were randomly selected in the sections, and the number of GSDMD-positive cells was counted. Cells were counted by two observers blinded to the identity of the mice, and the average number of RGCs was calculated from at least three independent biological replicates.

### 4.9. Western Blot

Western blotting was performed as previously described [22]. Mice were anesthetized at different time points after permanent MCAO-induced retinal ischemia to extract retinal tissue. Retinal tissue is homogenized to extract total protein. Primary RGC were harvested for protein extraction after the different treatments. The following antibodies were used: rabbit anti-Homer1 (1:1000; Abcam, Cambridge, UK), rabbit anti-TXNIP (1:1000; Affinity, Jiangsu, China), rabbit anti-NLRP3 (1:1000; CST, Danvers, MA, USA), mouse anti-cleaved capase1 (1:10,000; CST, Danvers, MA, USA), rabbit anti-GSDMD-N (1:1000; Affinity, Jiangsu, China), rabbit anti-Phospho-AMPK (1:1000; CST, Danvers, MA, USA), rabbit anti-AMPK (1:1000; CST, Danvers, MA, USA), rabbit anti-Phospho-LKB1 (1:1000; CST, Danvers, MA, USA), rabbit anti-LKB1 (1:1000; CST, Danvers, MA, USA), rabbit anti-IL-18 (1:1000; Proteintech, Wuhan, China), rabbit anti-IL-1β (1:1000; Proteintech, Wuhan, China), rabbit anti-Bip (1:1000; Abcam, Cambridge, UK), rabbit anti-CHOP (1:1000; Abcam, Cambridge, UK), and goat anti-mouse/rabbit secondary antibody (1:10,000; Abcam, Cambridge, UK). Images were acquired with a ChemiDoc imaging system (Bio-Rad, Hercules, CA, USA).

### 4.10. qPCR

Western blotting was performed as previously described [22]. Primary RGCs were harvested for RNA extraction after different treatments using the TRIzol reagent. Mice were anesthetized after permanent MCAO-induced retinal ischemia, and the retinal tissue was used for qPCR. Reverse transcription was performed according to the protocol of the HiScript II Q Select RT SuperMix for qPCR (+gDNA wiper) kit (Vazyme, Nanjing, China). qPCR was performed according to the protocol of the ChamQ SYBR Color qPCR Master Mix (Low ROX Premixed) kit (Vazyme, Nanjing, China). The primers for mRNA were as follows: Homer1-F: “AATGGTTAGGGGGCACTGTTT”, Homer1-R: “CCCATCTGCCACAGT CACAA”; and GAPDH-F: “AGAGGCCCTATCCCAACTCG, GAPDH-R: “GTGGGT GCAGCGAACTTTATT”. The expression of related RNAs was calculated using the 2^−ΔΔCt^ method, and GAPDH was used as a control. The experiment was repeated thrice.

### 4.11. Flash Visually Evoked Potential Recordings (FVEPs)

FVEPs were performed as previously described [49]. The C57BL/6 mice were dark-adapted overnight and then anesthetized with pentobarbital sodium (80 mg/kg) intraperitoneally. FVEPs were measured using a visual electrodiagnostic system. The body temperature of the mouse was maintained at 37 °C during the experiments. Three silver electrodes were implanted supraperiosteally at each hemisphere V1 region, the midpoint of the binoculars, and the tail of the mice. When checking the right eye, the left eye was covered by a black eye mask. White flash stimuli were shown at a frequency of 2 Hz for 250 ms. An average of 64 sweeps were collected, and the raw data were saved for further analysis. The P2 amplitude was measured to check visual function. The average of the three successive measurements at different consecutive time points of each mouse was calculated.

### 4.12. HE

The IPL thickness was measured in four adjacent areas within 1 mm of the optic disc. Firstly, the frozen sections were immersed in hematoxylin for 3 min, washed with running water for 10 s, and the eosin solution was immersed for 45 s. Then, the gradient ethanol solution was dehydrated, xylene was transparent, and a cover glass was covered. ImageJ software (Version 1.8.0, National Institutes of Health, Bethesda, MD, USA) was used to label the thickness [50]. Total retinal thickness (%) = retinal thickness in the experiment group/retinal thickness in the control group × 100.

### 4.13. Data Analysis

The statistical analysis was performed using GraphPad Prism software (version 9.0, San Diego, CA, USA). All experiments were performed three or more times unless otherwise indicated. Parametric and nonparametric tests were used according to the homogeneity of variance. The *t*-test was used to compare the mean of variance homogeneous data, and the rank sum test was used for non-homogeneous data. Multiple group comparisons of the means were performed by one-way analysis of variance (ANOVA) or the Student–Newman–Keuls (SNK) test. Differences between the groups were determined using ANOVA with Bonferroni post hoc test. *p* < 0.05 were statistically significant.

## Figures and Tables

**Figure 1 ijms-24-16811-f001:**
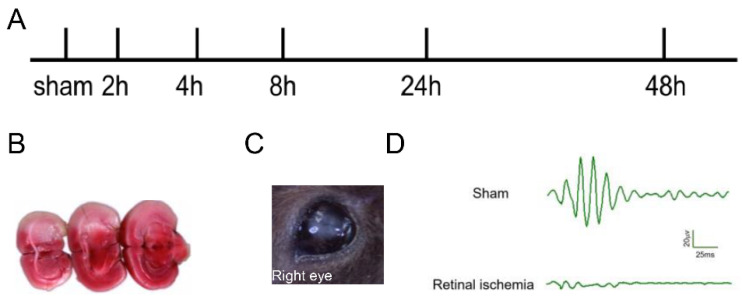
Generation of a MCAO-induced retinal ischemia model. (**A**) Experimental timeline design for present work. (**B**) Representative photographs of TTC-stained sections of right hemisphere ischemia induced by the MCAO model. (**C**) Representative photographs of “Whitening” following MCAO model-induced right retinal ischemia. (**D**) Oscillating potential of the right retina.

**Figure 2 ijms-24-16811-f002:**
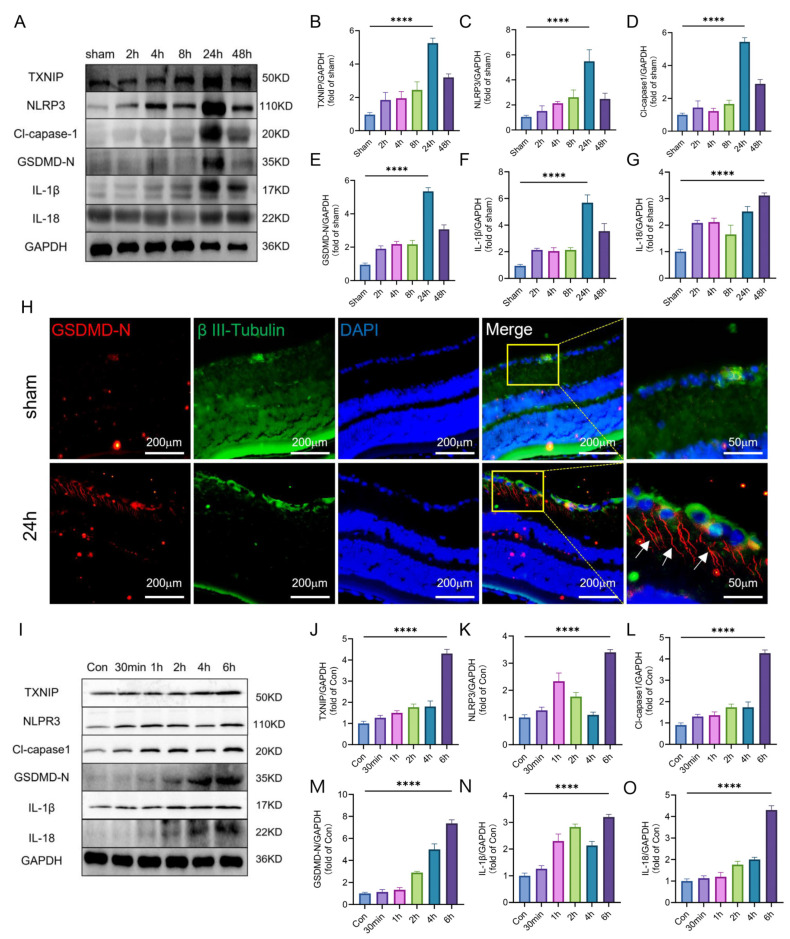
Time-Course of TXNIP/NLRP3-mediated pyroptosis. (**A**) TXNIP/NLRP3-mediated pyroptosis expression of the ischemic retina in vivo. The blots are representative of other replicates in those groups. (**B**–**G**) Quantification of result in panel (**A**) (n = 6 for each group). (**H**) Representative photographs of GSDMD and Neuronal β-III Tubulin co-staining of frozen sections of retina tissue at 24 h after MCAO-induced right retinal ischemia (white arrows show axons that undergo pyroptosis). (**I**) Representative immunoblot of TXNIP/NLRP3-mediated pyroptosis expression in cultured primary RGC after OGD in different times; Con: control group, RGCs cultured in normal Neurobasal. (**J**–**O**) Quantification of result in panel I (n = 3 for each group). Data are expressed as the mean ± SD. For panel (**B**–**G**,**J**–**O**): **** *p* < 0.0001 by one-way-ANOVA analysis. All data are representative of three independent experiments.

**Figure 3 ijms-24-16811-f003:**
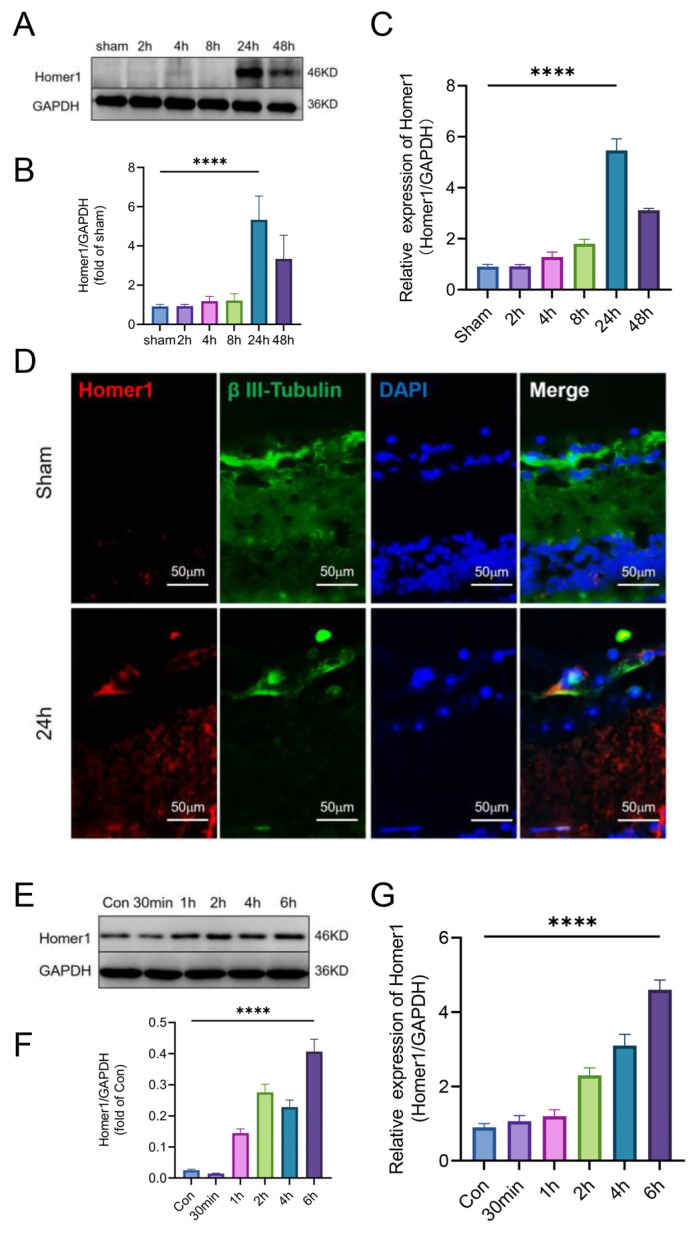
Protein and mRNA expression levels of Homer1 after MCAO model-induced right retinal ischemia. (**A**) Homer1 protein expression of the ischemic retina in vivo. The blots are representative of other replicates in those groups. (**B**) Quantification of result in panel (**A**) (n = 6 for each group). (**C**) Homer1 mRNA expression in vivo after MCAO model-induced right retinal ischemia (n = 6 for each group). (**D**) Representative photographs of Homer1 and Neuronal β-III Tubulin co-staining of frozen sections of the retina at 24 h after MCAO-induced right retinal ischemia. (**E**) Protein expression of Homer1 in primary RGCs after OGD. (**F**) Quantification of result in panel (**E**) (n = 3 for each group). (**G**) mRNA level of Homer1 (n = 3 for each group). Data are expressed as the mean ± SD. For panel (**B**,**C**,**F**,**G**): **** *p* < 0.0001 by one-way-ANOVA analysis. All data are representative of three independent experiments.

**Figure 4 ijms-24-16811-f004:**
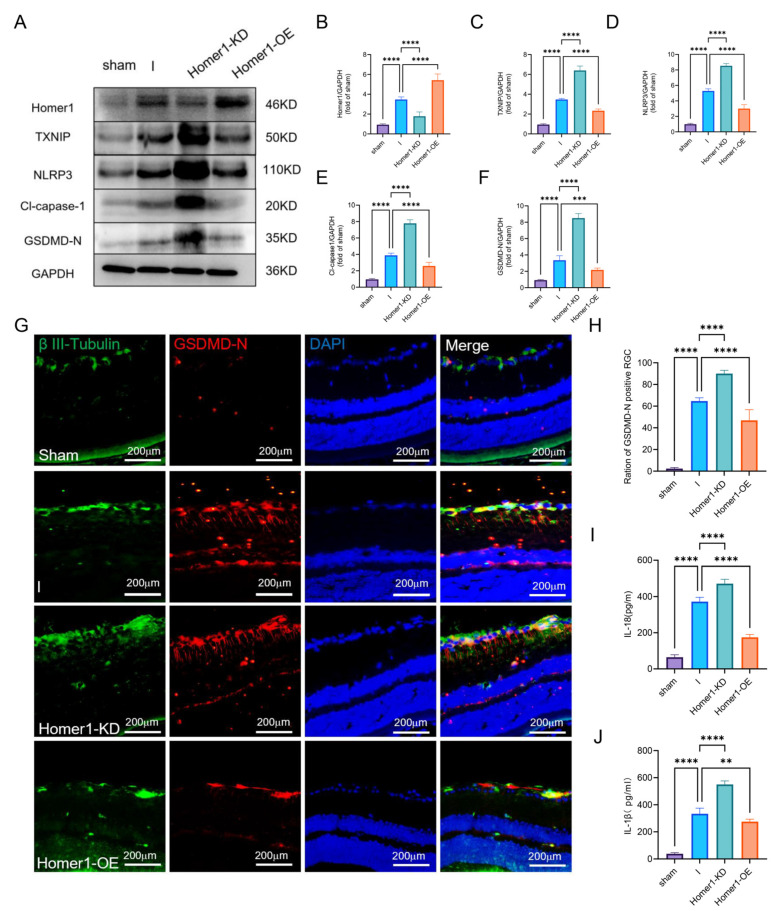
Overexpression of Homer1 alleviated TXNIP/NLRP3-mediated pyroptosis in vivo. (**A**) Effects of Homer1-OE and Homer1-KD on the expression level of TXNIP/NLRP3-mediated pyroptosis-related proteins were examined by Western blot in vivo at 24 h after retinal ischemia. (**B**–**F**) Quantification of result in panel (**A**) (n = 6 for each group). (**G**) Representative photographs of GSDMD-positive RGC in each group. (**H**) Quantification of result in panel (**G**) (n = 6 for each group). (**I**) Expression of IL-18 (n = 6 for each group). (**J**) Expression of IL-1β (n = 6 for each group). Data are expressed as the mean ± SD. For panel (**B**–**F**,**H**–**J**): ** *p* < 0.01, *** *p* < 0.001, and **** *p* < 0.0001 by one-way-ANOVA analysis. All data are representative of three independent experiments.

**Figure 5 ijms-24-16811-f005:**
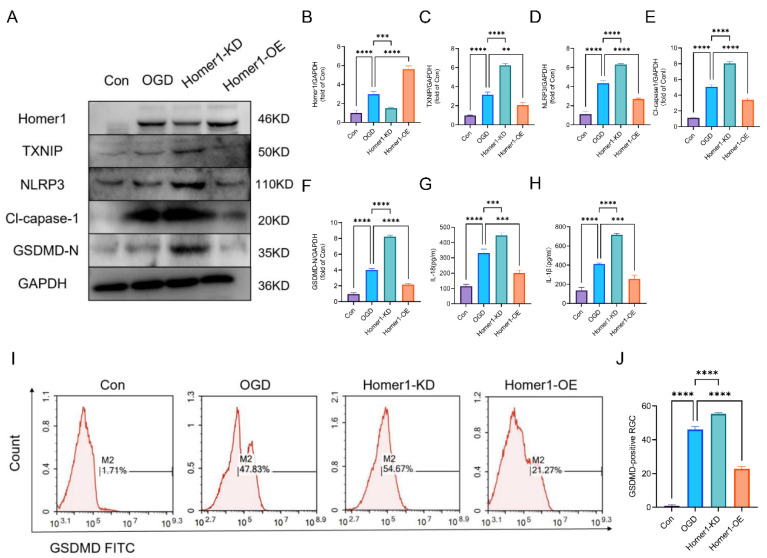
Overexpression of Homer1 alleviated TXNIP/NLRP3-mediated pyroptosis in RGC cells in vitro. (**A**) Effects of Homer1-OE and Homer1-KD on the expression level of TXNIP/NLRP3-mediated pyroptosis-related proteins were examined by Western blot in vitro at 6 h after OGD. (**B**–**F**) Quantification of result in panel (**A**) (n = 3 for each group). (**G**) Expression of IL-18 (n = 3 for each group). (**H**) Expression of IL-1β (n = 3 for each group). (**I**) and quantification (**J**) of GSDMD-positive RGC cells in different groups measured in vitro based on flow cytometry at 24 h after OGD (n = 3 for each group). Data are expressed as the mean ± SD. For panel (**B**–**F**,**H**): ** *p* < 0.01, *** *p* < 0.001, and **** *p* < 0.0001 by one-way-ANOVA analysis. All data are representative of three independent experiments.

**Figure 6 ijms-24-16811-f006:**
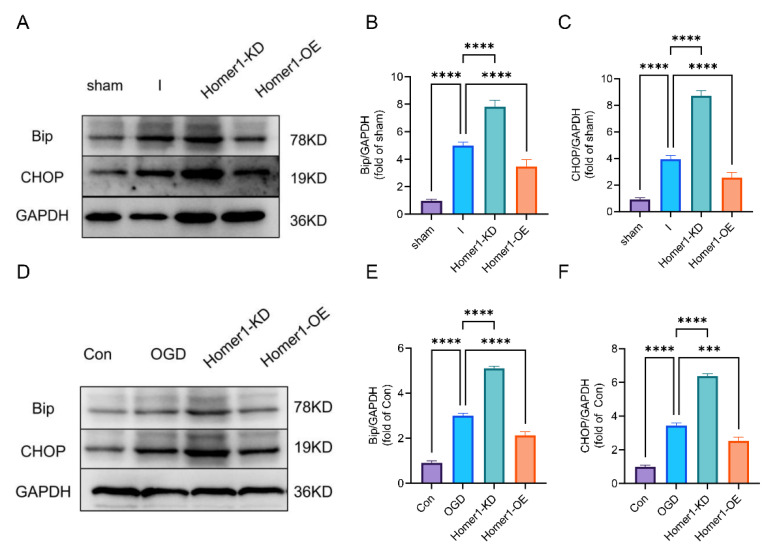
Overexpression of Homer1-alleviated endoplasmic reticulum stress. (**A**) Expression levels of Bip and CHOP protein were tested by Western blotting measurement in vivo at 24 h after retinal ischemia. (**B**,**C**) Quantification of result in panel (**A**) (n = 6 for each group). (**D**) Expression levels of Bip and CHOP protein were tested by Western blot measurement in vitro at 6 h after OGD. (**E**,**F**) Quantification of result in panel (**C**) (n = 3 for each group). Data are expressed as the mean ± SD. For panel (**B**,**C**,**E**,**F**): *** *p* < 0.001 and **** *p* < 0.0001 by one-way-ANOVA analysis. All data are representative of three independent experiments.

**Figure 7 ijms-24-16811-f007:**
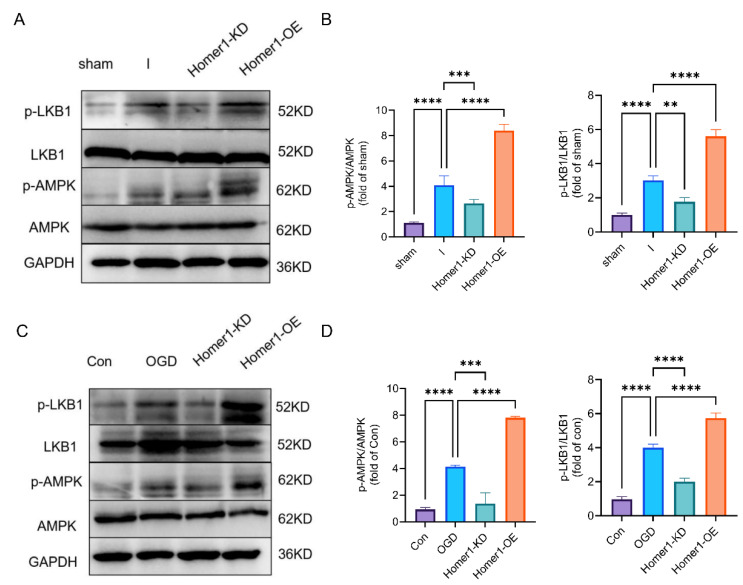
Effects of Home1-OE and Homer-KD on upstream kinase LKB1 and AMPK signaling. (**A**) WB was performed to examine the effects of Homer1-OE and Homer1-KD on upstream kinase LKB1 and AMPK signaling protein expression in vivo at 24 h after retinal ischemia. (**B**) Quantification of result in panel (**A**) (n = 6 for each group). (**C**) WB was performed to examine the effects of Homer1-OE and Homer1-KD on upstream kinase LKB1 and AMPK signaling protein expression in vitro at 6 h after OGD. (**D**) Quantification of result in panel (**C**) (n = 3 for each group). Data are expressed as the mean ± SD. For panel (**B**,**D**): ** *p* < 0.01, *** *p* < 0.001 and **** *p* < 0.0001 by one-way-ANOVA analysis. All data are representative of three independent experiments.

**Figure 8 ijms-24-16811-f008:**
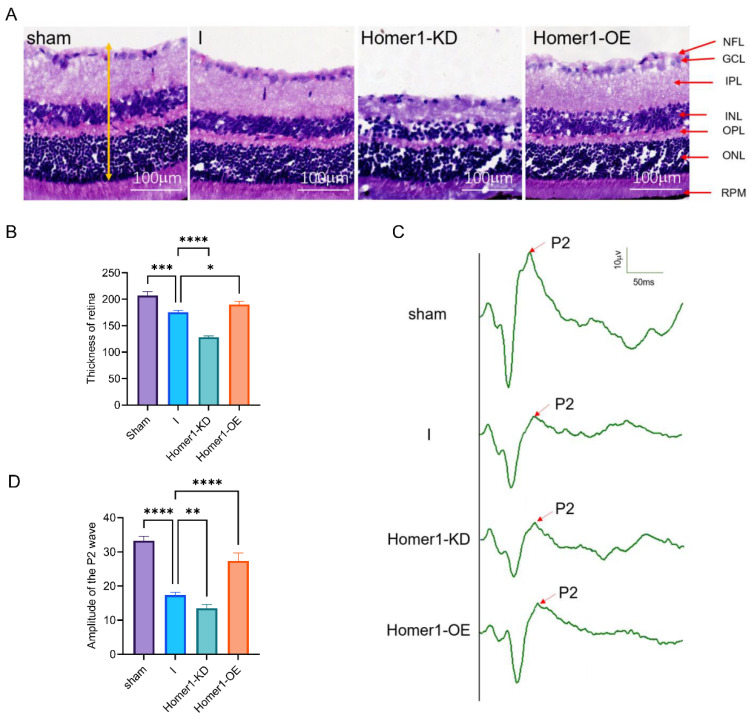
Overexpression of Homer1-attenuated ischemic injury of the retina. (**A**) Hematoxylin and eosin staining of retinal cross-sections showed the decreased thickness of retinal tissue (marked by the yellow arrow) 24 h after MCAO model-induced right retinal ischemia, and this effect was markedly inhibited with Homer1-OE treatment. Representative images from the sham group, I group, Homer1-OE group, and Homer1-KD group. (**B**) Retinal tissue thickness was assessed by H&E staining and Image J analysis (n = 6 for each group). (**C**) Representative Flash visual–evoked potential (FVEP) of right eye profile in each group. (**D**) Quantification of the amplitude of the P2 wave in panel (**C**) (n = 6 for each group). Data are expressed as the mean ± SD. For panel (**B**,**D**): * *p* < 0.05, ** *p* < 0.01, *** *p* < 0.001, **** *p* < 0.0001 by one-way-ANOVA analysis. All data are representative of three independent experiments. NFL: nerve fiber layer; GCL: ganglion cell layer; IPL: inner plexiform layer; INL: inner nuclear layer; OPL: outer plexiform layer; ONL: outer nuclear layer; RPE: retinal pigment epithelium.

## Data Availability

The datasets used and/or analyzed during the current study are available from the corresponding author upon reasonable request.

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
