# Peer review of "Homer1 Protects against Retinal Ganglion Cell Pyroptosis by Inhibiting Endoplasmic Reticulum Stress-Associated TXNIP/NLRP3 Inflammasome Activation after Middle Cerebral Artery Occlusion-Induced Retinal Ischemia"

_ijms, 2023, doi:10.3390/ijms242316811_

Round 1

Reviewer 1 Report

Comments and Suggestions for Authors

The manuscript “Homer1 protects against retinal ganglion cell pyroptosis by inhibiting
endoplasmic reticulum stress-associated TXNIP/NLRP3 inflammasome activation
after MCAO-induced retinal ischemia” is well written and explores the hither to unknown function of Homer1 in MCAO-induced retinal ischemia. The conclusions of the study are mostly well-supported by the data; however, certain aspects could be improved to strengthen the findings.

1.      Introduction (line 45, first sentence and 2nd sentence)…First and second sentences are too vague and are not supported by solid references.

2.      Suggest authors to use the term “generation” of MCAO-induced retinal ischemia model…instead of “construction” of MCAO-induced retinal ischemia model….the word construction is misleading. Suggest authors to remove the term construction (and instead use the term “generation”…all through the paragraph.

3.      Section 2.2 line 96 the vague reference to “pyroptosis –related protein” is too generalized and request authors to include the names of specific proteins in the main text (that were studied using Western blot).

4.      Section 2.2 line 101…authors claimed that GSDMD-N expression was also present in axons…Did the authors stain for axons as they did for RGC cells?...or is it just a morphological observation? If that is the case then authors should include pics showing axonal localization of GSDMD-N.

5.      Section 2.3 line 118…authors used the term “spatiotemporal”,….while the authors showed temporal (time dependent) increase of Homer1….they haven’t shown spatial upregulation of Homer1…suggest authors to remove the word spatiotemporal and instead use ‘time dependent increase or temporal induction of Homer1.

6.      Section 2.3 line 118…authors should elaborate the abbreviation WB …use Western blot..

7.      Similarly authors should elaborate the abbreviation OGD (Oxygen glucose deprivation) and I group (Ischemia group) in the manuscript.

8.      Fig 2 I (and also in various figures of in vitro RGC culture)…authors should clarify what “con” means…is it a media control? Or a time point control (0 hrs)?

9.      It would have been great if authors show.. how Homer1 regulate AMPK phosphorylation? Does Homer1 function as an adapter molecule mediating AMPK phosphorylation with an upstream kinase? Does Homer1 interact physically with AMPK?

Comments on the Quality of English Language

English language can be improved for meaningful interpretation

Reviewer 2 Report

Comments and Suggestions for Authors

This paper investigates the role of the Homer1 protein after induced retinal ischemia in mice. Homer1 has anti-inflammatory properties and its overexpression not only attenuates retinal ganglion cell pyroptosis (a special form of cell death), but also inhibits the release of pro-inflammatory factors and ER stress. The authors concluded that Homer1 exerts a protective function against retinal ischemia induced by middle cerebral artery occlusion (MCAO).

The content of the paper is interesting and the study is also well written, but there are some points of criticism that should be considered by the authors.

Major comments

Fig. 8A. This figure is not convincing. The images of the HE stained sections are out of focus and show many artifacts (holes) questioning the conclusions of the authors. In addition, a scale bar is missing. The authors should label the layer of the ganglion cells. What was the thickness of the sections? From the images, it appears that the thickness of the outer nuclear layer of the retina has increased in the Homer-1 group. Do the authors have any explanation for this?

Do the authors have an explanation for why only ganglion cells and no other cells of the retina are affected after retinal ischemia?

Minor comments

Line (L) 64: it should read endoplasmic reticulum (ER) stress

L 81: explain TTC staining

L 101: explain Con

Round 2

Reviewer 2 Report

Comments and Suggestions for Authors

In the revised version of the manuscript, the authors have satisfactorily addressed my points of criticism. I have no further comments.